# Metastatic Lymph Node Ratio (mLNR) is a Useful Parameter in the Prognosis of Colorectal Cancer; A Meta-Analysis for the Prognostic Role of mLNR

**DOI:** 10.3390/medicina55100673

**Published:** 2019-10-04

**Authors:** Jung Soo Pyo, Joo Heon Kim, Seung Yun Lee, Tae Hwa Baek, Dong Wook Kang

**Affiliations:** 1Department of Pathology, Eulji University Hospital, Eulji University School of Medicine, Daejeon 35233, Korea; jspyo@eulji.ac.kr (J.S.P.); kjh2000@eulji.ac.kr (J.H.K.); sylee@eulji.ac.kr (S.Y.L.); 2Study Group for Meta-Analysis, Eulji University Hospital, Eulji University School of Medicine, Daejeon 35233, Korea; 3Medical Examiner’s office, National Forensic Service, Wonju 26460, Korea; komsol20th@hanmail.net

**Keywords:** metastatic lymph node ratio, colorectal cancer, meta-analysis, prognostic factor

## Abstract

*Background and objectives:* The presenting study aimed to elucidate the prognostic role of the metastatic lymph node ratio (mLNR) in patients with colorectal cancer (CRC), using a meta-analysis. *Materials and Methods:* Using data from 90,274 patients from 14 eligible studies, we performed a meta-analysis for the correlation between mLNR and survival rate. Besides, subgroup analyses were performed, based on tumor stage, tumor location, and mLNR. *Results:* A high mLNR showed significant correlation with worse overall survival and disease-free survival rates in CRC patients (hazard ratio (HR), 1.617, 95% confidence interval (CI) 1.393–1.877, and HR 2.345, 95% CI 1.879–2.926, respectively). In patients with stage III, who had regional LN metastasis, the HRs were 1.730 (95% CI 1.266–2.362) and 2.451 (95% CI 1.719–3.494) for overall and disease-free survival, respectively. According to tumor location, rectal cancer showed a worse survival rate when compared to colon cancer. In the analysis for overall survival, when mLNR was 0.2, HR was the highest across the different subgroups (HR 5.040, 95% CI 1.780–14.270). However, in the analysis for disease-free survival, the subgroup with an mLNR < 0.2 had a higher HR than the other subgroups (HR 2.878, 95% CI 1.401–5.912). *Conclusions:* The mLNR may be a useful prognostic factor for patients with CRC, regardless of the tumor stage or tumor location. Further studies are necessary for the detailed criteria of mLNR before its application in daily practice.

## 1. Introduction

Colorectal cancer (CRC) is representing about 10% of all new cancers, and the mortality rates are variable according to tumor stage or treatment availability [1]. In the American Joint Committee on Cancer (AJCC) cancer staging, the nodal disease, N stage, is defined based on the presence of regional lymph node (LN) metastasis [2]. The assessment for lymph node metastasis in colorectal cancer is performed via several affected regional LNs according to the 8th AJCC cancer staging manual. [2]. In the current staging system, CRC with regional LN metastasis classifies into stage III, which has additional treatment options, such as adjuvant chemotherapy [3]. Although tumor staging is useful for prediction of prognosis, the prognosis between CRC with stage IIIA and II is not clearly separated [4,5]. A more detailed evaluation system for the nodal disease will be required.

In the National Comprehensive Cancer Network (NCCN) guidelines, for proper evaluation of nodal disease, minimal requirement of examination of regional LN is recommended as 12 harvested LNs [6]. Inadequate harvest of regional LN may lead to the false-negative nodal disease or lower N stage [4]. The compensation for these possibilities could be needed to evaluate the nodal disease in cancer staging. Other parameters using nodal status have been introduced, such as harvested number of LN, the number of metastatic LN (mLN), and metastatic LN ratio (mLNR) [7,8,9,10,11,12,13,14,15,16,17,18,19,20].

The mLNR defines as the ratio of the number of mLNs to the number of examined LNs. In the present study, using a meta-analysis, the usefulness of mLNR for prediction of prognosis was evaluated in CRCs. Also, to obtain the optimal criteria of mLNR for predicting worse prognosis of CRCs, the comparison between various criteria of mLNR was performed from eligible studies.

## 2. Materials and Methods

### 2.1. Searching Studies and Selection Criteria

Relevant published articles were identified by searching the PubMed and MEDLINE databases through to 30 January 2019. The meta-analysis was searched using the following keywords: “colon OR rectum OR colorectal” AND “lymph node ratio OR metastatic lymph node ratio.” All retrieved titles and abstracts of the potentially eligible studies were reviewed for the exclusion of the study by two independent authors. Articles were selected if the study was performed in human CRCs and if there was a correlation between mLNR and survival rate of CRC. Review articles concerning the experimental study were also screened to find eligible studies. We excluded articles if they were non-original articles or case reports, or if the article was published in a language other than English. In addition, the assessment of the quality of the eligible studies was performed using the Newcastle-Ottawa Scale [21].

### 2.2. Data Extraction

Two independent authors extracted data from all eligible studies. The selected data were obtained from each of the eligible studies [7,8,9,10,11,12,13,14,15,16,17,18,19,20]: We collected the author’s first name, the publication year of articles, the location of the research, number of patients analyzed, and the correlation between high mLNR and survival rates. The correlation between high mLNR and survival outcomes was evaluated according to the hazard ratio (HR) for the quantitative aggregation of survival results. If the studies were not quoting the HR and confidence interval (CI), the survival rates were obtained from the presented data using the log-rank statistics and its *p* value, and the O-E statistic (the difference between observed and expected events) or its variance. If the survival data were unavailable, HR was measured using the number of patients, the total events at high risk, and the log-rank statistic and its *p* value in each group. Finally, if the useful survival data were in the form of graphics or other visualized representations, clinical survival outcomes were extracted at specific time-point to calculate the HR estimate and its variance under the expectation that patients developed at a constant rate during the time intervals. The clinical survival rates were obtained by two independent authors to reduce variability. The HRs were then integrated into an overall HR using the Peto odds ratio method [22].

### 2.3. Statistical Analyses

We analyzed all survival data using the Comprehensive Meta-Analysis software package (Ver. 2, Biostat, Englewood, NJ, USA) to perform the meta-analysis. For the meta-analysis, we investigated the survival rates of high and low mLNR in CRC and performed subgroup analysis, based on tumor stage, tumor arising location, and criteria of mLNR. We also checked heterogeneity between the data of the selected studies by the *Q*-value and *I*^2^ statistics and expressed as *p* values. Additionally, a sensitivity analysis was checked to assess the heterogeneity of available studies and the impact of each study on the combined effect. In addition, the statistical significance of the difference between subgroups was evaluated using a meta-regression test. In the current study, because available studies used various or different diagnostic criteria and population, the random-effect model rather than the fixed-effect model was more suitable to perform the study. Begg’s funnel plot and Egger’s test were primarily used for the assessment of publication bias. If notable publication bias was presented, the fail-safe N and trim-fill tests were additionally conducted to confirm the degree of publication bias. We considered statistically significant at *p* < 0.05 in all results of the present study.

## 3. Results

### 3.1. Selection and Characteristics of the Studies

Four hundred three articles were identified in the PubMed and MEDLINE databases search. Among them, 291 studies were excluded due to insufficient or no information. Other studies excluded because they involved other diseases (*n* = 52), non-English (*n* = 18), use of animals or cell lines (*n* = 11), and were non-original articles (*n* = 17). Finally, 14 studies were included in this systematic review and meta-analysis (Figure 1).

These studies included 90,274 CRC patients from 14 eligible studies. All studies were retrospectively investigated, and three studies used the local cancer registry [9], multi-institutional database [19], and cancer registry database [7]. Most studies showed high scorings in the Newcastle-Ottawa Scale except one study (Appendix A). Main characteristics of eligible studies were summarized in Table 1.

### 3.2. Meta-Analysis

The estimated mean number of harvested lymph nodes was 23.365 (95% CI 20.993–25.736) in overall cases. In subgroups by tumor location, the estimated mean number of harvested lymph nodes was 27.709 (95% CI 24.435–30.983) and 15.511 (95% CI 10.219–20.803) in colon and rectum, respectively. The harvested number of lymph nodes was significantly higher in the colon than in the rectum (*p* = 0.001 in a meta-regression test). In the present study, we performed the meta-analysis divided into overall and disease-free survival rates. The high mLNR was significantly correlated with worse overall, and disease-free survival (HR 1.617, 95% CI 1.393–1.877 and HR 2.345, 95% CI 1.879–2.926, respectively, Figure 2). The impact of Zhang’s report (2018) might be larger than other eligible studies. The impact of the individual study was evaluated by a sensitivity analysis. In omitting Zhang’s report, the estimated HR was 1.834 (95% CI 1.391–2.419). Compared to overall cases (HR 1.617, 95% CI 1.393–1.877), the tendency did not differ. Next, subgroup analysis was conducted based on tumor stage, tumor location, and the criteria of mLNR. Firstly, the correlation between high mLNR and overall survival rate was investigated. In subgroup analysis based on tumor stage, HR of subgroup included all stage tumors was 2.420 (95% CI 1.141–5.129). However, in the subgroup with only stage III tumor, HR was 1.730 (95% CI 1.266–2.362). Among CRCs, HR of rectal cancer was higher than that of colon cancer (5.040, 95% CI 1.780–14.270 vs. 1.785, 95% CI 1.213–2.626). In disease-free survival rate, HR of CRC with stage III was lower than CRC with all tumor stages, and HR of rectal cancers was higher than colon cancers, unlike overall survival rate. 

In eligible studies, the criteria of mLNR ranged from 0.125 to 0.30, and the median value of criteria was 0.19. In the present study, based on mLNR 0.20, eligible studies divided into high and low mLNRs criteria subgroups. In the overall survival rate, when mLNR was 0.2, the HR was highest among various criteria subgroups (HR 5.040, 95% CI 1.780–14.270, Figure 3 and Table 2). However, in the subgroup with mLNR < 0.2, the HR was lowest as 1.521 (95% CI 1.394–1.661) (Table 2). In the disease-free survival rate, when mLNR was 0.2, the HR was 2.425 (95% CI 1.485–3.960). However, unlike the overall survival rate, the subgroup with mLNR < 0.2 was showed higher HR than other subgroups (Table 3). Also, in the meta-regression test, there was no significant difference between subgroups of tumor stage, tumor location, and mLNR value.

## 4. Discussion

The most useful prognostic factor for CRCs is an anatomic extension of CRC [2]. This anatomic extension is evaluated by tumor stage, such as AJCC cancer staging [2]. CRCs with LN metastasis are classified into stage III in the current edition of AJCC cancer staging [2]. However, regardless of the harvested regional LN number, node (N) stage is affected by only metastatic LN number. Recently, to compensate for the limitation of the N stage, mLNR has been introduced and studied. The present study is the first meta-analysis of the prognostic role of mLNR in the CRCs.

Patients with stage III CRC have several therapeutic options, such as adjuvant radiotherapy, either alone or concurrent chemoradiotherapy [2,3]. According to the recent 8th edition of AJCC cancer staging, CRCs with LN metastasis define as stage III [2]. So, the detection of nodal disease may be important for the decision of treatment modality and prediction of prognosis. The detection of nodal disease can be affected by various factors, including surgical, pathologic, and tumor status [23,24,25,26]. Baxter et al., reported that only 37% of CRC patients performed adequate LN evaluation in a population-based analysis [24]. Also, the number of harvested LN after surgical resection was correlated with survival of the CRC patients [25,27]. The current guidelines recommend examining minimally 12 regional LNs from the surgical specimen to evaluate the nodal disease [28]. However, the number of harvested LN may also be affected by modifiable factors, such as the adequacy of the surgical resection and the proper examining pathologists [2,28]. Therefore, the proper detection of nodal disease will be guaranteed from the proper and careful examination. Indeed, the prognostic effect of harvested LN number should be considered in the interpretation of the prognostic role of nodal disease.

When the classification system for prognosis is powerful, the prognosis can be clearly separated by the classification system alone. However, CRC with stage IIIA, which is confined to submucosa or proper muscle with the nodal disease of N1/N1c or N2a stage, has favorable prognosis compared to CRC with stage II [2]. In AJCC cancer staging, N stage is decided by only affected regional LNs in CRC. That is, a more detailed evaluation system for the nodal disease will be needed. The prognostic separation between CRC with and without the nodal disease (N1/2 vs. N0) was found in the previous study [29]. Previously, the correlation between the number of harvested LN and prognosis of CRC has been studied [25,27]. If harvested LN number was less than 10, there was no significant correlation between the number of harvested LN and prognosis [20]. Indeed, the number of affected LN may be increased by the number of harvested LN [20]. If the harvested LN can affect on N stage, the consideration of the number of harvested LN can be important in the evaluation of the N stage. However, in the present AJCC cancer staging, harvested number of LN does not affect the evaluation of N stage [2]. To compensate for the limitation, the prognostic role of mLNR in colon cancer with stage III was first investigated by Berger et al. [30]. To evaluate the nodal disease, the minimal requirement of harvested LN number is needed, and the required number of harvested LN is twelve [2]. If the minimum requirement is twelve for proper evaluation of nodal disease, 1.5–3.6 affected LNs are needed as defining of high mLNR in eligible studies according to the high mLNR criteria (0.125–0.3). As a result, based on harvested LNs, the nodal disease will be decided as N1 (affected LN 1–3) or N2 (affected LN > 4). However, the previous study reported that there was no interaction between mLNR and AJCC N categorization [20]. That is, high mLNR was not correlated with N stage of AJCC cancer staging. In CRCs with the same affected LN number, the difference of prognosis according to harvested LN number is not fully understood.

The studies for mLNR have been reported in various malignant tumors, such as gastric, pancreatic, and breast cancers [31,32,33,34,35]. Eligible studies of the present meta-analysis with criteria for high mLNR were ranged from 0.125 to 0.3. The criteria for high mLNR in eligible studies were defined by independent guidelines [7,8,9,10,11,12,13,14,15,16,17,18,19,20]. Also, each study included CRCs with various tumor stages and population. Because direct comparison between criteria for high mLNR is impossible, a meta-analysis can be useful for evaluation of optimal criteria for high mLNR. Interestingly, in subgroup analysis based on criteria for high mLNR, the inverse correlation between overall and disease-free survival rates was found (Table 2 and Table 3). In our results, according to the inclusion of criteria 0.2, HRs differed between subgroups. In the results of ratio-based classification according to the mLNR gradient, high mLNR was significantly correlated with lower survival rate [30,33]. However, in our result, high mLNR criteria showed lower HR in disease-free survival compared to low mLNR criteria. Some researchers suggested that minimal 15 harvested LNs are needed for evaluation using the ratio-based system [30]. However, the impact of harvested LN number on prognosis cannot be excluded in the evaluation of the prognostic role of mLNR. Although the mLNR can be useful as the powerful predictor for prognosis of CRCs, the conclusive information for criteria is not fully elucidated in the current meta-analysis. Obviously, high mLNR was significantly correlated with worse prognosis in CRCs. To confirm the optimal criteria for high mLNR, further cumulative will be needed.

There are some limitations in the current meta-analysis. First, some reports have demonstrated the correlation between mLNR and survival rate divided by several gradients, but no high and low mLNR [12,14,30]. However, those reports were not included in the current meta-analysis. Second, in the previous study, mLNR had an independent prognostic role, regardless of the harvested LN number [20]. Because the impact of the harvested LN number on mLNR was not investigated in the current study, cumulative studies will be needed. Besides, the minimal requirement of the harvested LN number for proper evaluation of mLNR will be needed before the application of mLNR. Third, in the subgroup with high mLNR criteria, the HRs differed between overall and disease-free survival rates. However, the reason could not be found in the current study. Fourth, among eligible studies, some reported the results from less than twelve harvested LNs. In eligible studies, the average of harvested LNs was from 7 to 29.4. So, the merge of the reported prognostic roles from each eligible study is difficult. Fifth, the present study could not deal with the detailed analysis based on tumor grade, histologic subtype, total mesorectal excision operation, or preoperative adjuvant therapy because of insufficient information of eligible studies. Sixth, the value of mLNR can be higher in cases with aggressive behaviors than in cases with indolent behaviors. However, the detailed analysis could not be performed due to insufficient information on eligible studies.

## 5. Conclusions

The present study is the first meta-analysis of the prognostic role of mLNR in CRCs. In conclusion, our data showed that high mLNR was significantly correlated with worse overall and disease-free survival rates. Besides, regardless of tumor location, mLNR is useful for prediction of prognosis in CRCs. For evaluation of optimal criteria of mLNR in CRCs, further detailed studies will be required.

## Figures and Tables

**Figure 1 medicina-55-00673-f001:**
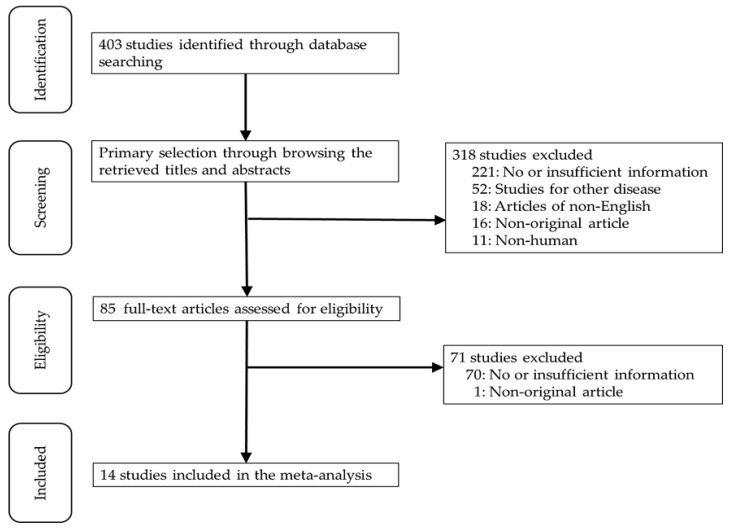
Flow chart of search study and selection methods.

**Figure 2 medicina-55-00673-f002:**
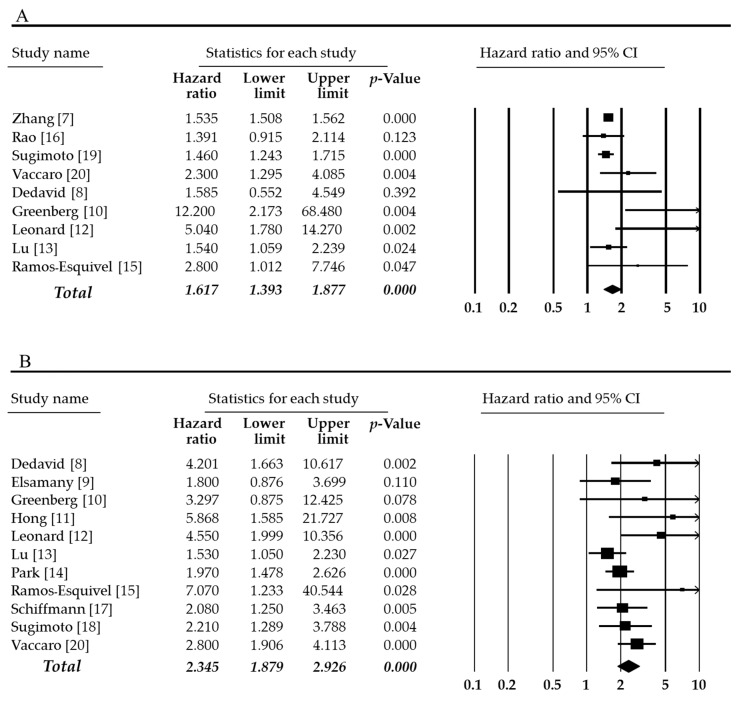
Forest plots for the correlations between high metastatic lymph node ratio and worse survival rates. (**A**) Overall survival and (**B**) disease-free survival. There was statistically significant when *p*-value was <0.05.

**Figure 3 medicina-55-00673-f003:**
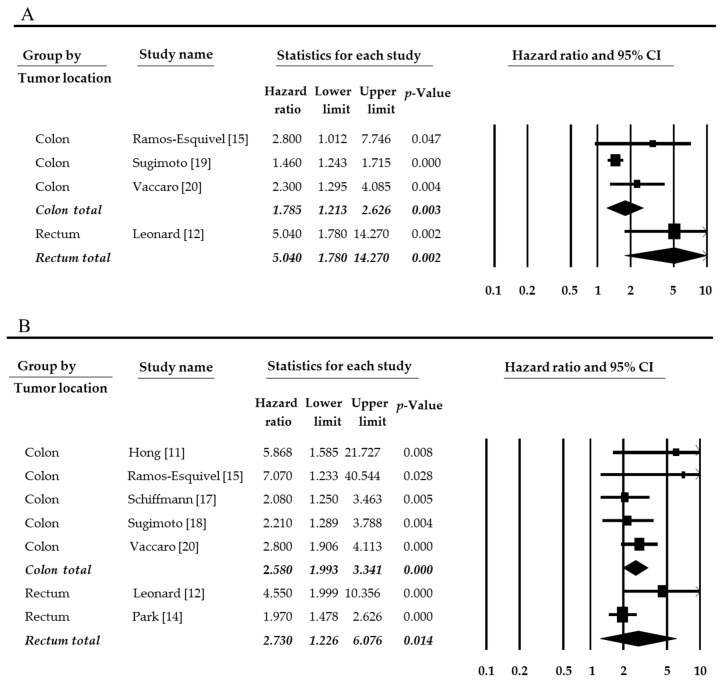
Forest plots for subgroup analysis by tumor location. (**A**) Overall survival and (**B**) disease-free survival. There was statistically significant when *p*-value was <0.05.

**Table 1 medicina-55-00673-t001:** Main characteristics of eligible studies.

Study, Year	Location	Study Design ^†^	Tumor Stage	Tumor Location	Age (SD)	Harvested Number of LN (SD)	mLNR Criteria	Number of Patients		
								LNM	High mLNR	Low mLNR
Dedavid, 2013 [8]	Brazil	R/SI	Stage III	Colorectum	62.4 (7.3)	19.3 (2.0)	0.15	70	32	38
Elsamany, 2014 [9]	Egypt	R/LC	Stage I–III	Colorectum	ND	ND	0.2	84	47	37
Greenberg, 2011 [10]	Israel	R/SI	Stage III	Colorectum	ND	20.3 (8.3)	0.13	65	35	30
Hong, 2011 [11]	Korea	R/SI	Stage III	Colon	ND	50.7 (21.7)	0.1638	130	33	97
Leonard, 2016 [12]	Belgium	R/SI	All stage (III, *n* = 145)	Rectum	65.6 (12.8)	12.8 (8.8)	0.2	357	ND	ND
Lu, 2013 [13]	Taiwan	R/SI	Stage III	Colorectum	64.9 (12.8)	19.8 (9.5)	0.17	612	322	290
Park, 2015 [14]	Korea	R/SI	Stage III	Rectum	54.0 (10.3)	18.2 (8.5)	0.25	724	ND	ND
Ramos-Esquivel, 2014 [15]	Costa Rica	R/SI	All stage (III, *n* = 29)	Colon	62.8 (17.7)	12.0 (2.7)	0.25	39	23	16
Rao, 2016 [16]	UK	R/SI	Duke C	Colorectum	72.0 (10.9)	20.1 (8.6)	0.125	147	81	66
Schiffmann, 2013 [17]	Germany	R/SI	Stage III	Colon	67.7 (9.5)	23.2 (9.3)	0.2	142	54	88
Sugimoto, 2013 [18]	Japan	R/SI	Stage III *	Colon	62.5 (10.0)	36.6 (17.3)	0.3	311	55	256
Sugimoto, 2015 [19]	Japan	R/MIDJ	Stage III	Colon	60.2 (12.2)	41.3 (20.2)	0.18	4172	1428	2744
Vaccaro, 2009 [20]	Argentina	R/SI	Stage III	Colon	67.4 (12.6)	20.0 (0.3)	0.25	362	92	270
Zhang, 2018 [7]	USA	R/SEER	Stage III	Colon	68.1 (13.5)	14.2 (9.6)	0.25	83,059	48,013	35,046

^†^ The study design is described as follows: R, retrospective; SI, single-institutional study; L, local cancer registry; MIDJ, multi-institutional database in Japan; SEER, surveillance, epidemiology, and end results cancer registry. SD, standard deviation; LN, lymph node; ND, No description; mLNR, metastatic lymph node ratio; LNM, lymph node metastasis.* because there was no information for patients with distant metastasis, the presence of stage IV cannot be excluded.

**Table 2 medicina-55-00673-t002:** Subgroup analysis for overall survival in colorectal cancer.

	Number of Subset	Fixed Effect (95% CI)	Heterogeneity Test (*p-*Value)	Random Effect (95% CI)	Egger’s Test (*p-*Value)
Overall	9	1.535 (1.509, 1.562)	0.072	1.617 (1.393, 1.877)	0.093
Tumor stage					
Stage III	5	1.536 (1.334, 1.769)	0.098	1.730 (1.266, 2.362)	0.108
High mLNR criteria					
mLNR < 0.2	5	1.534 (1.507, 1.561)	0.294	1.521 (1.394, 1.661)	0.601
mLNR ≤ 0.2	6	1.534 (1.508, 1.561)	0.084	1.557 (1.343, 1.805)	0.294
mLNR ≥ 0.2	3	2.770 (1.765, 4.347)	0.433	2.770 (1.765, 4.347)	0.398
Colon	2	2.412 (1.463, 3.977)	0.741	2.412 (1.463, 3.977)	NA
Rectum	1	5.040 (1.780, 14.270)	1.000	5.040 (1.780, 14.270)	NA

CI, confidence interval; mLNR, metastatic lymph node ratio; NA, not applicable. Colon cancer represent mLNR > 0.2, and rectal cancer present mLNR = 0.2, respectively.

**Table 3 medicina-55-00673-t003:** Subgroup analysis for disease-free survival in colorectal cancer.

	Number of Subset	Fixed Effect (95% CI)	Heterogeneity Test (*p-*Value)	Random Effect (95% CI)	Egger’s Test (*p-*Value)
Overall	11	2.209 (1.883, 2.591)	0.129	2.345 (1.879, 2.926)	0.020
Tumor stage					
Stage III	6	2.246 (1.795, 2.809)	0.091	2.451 (1.719, 3.494)	0.165
High mLNR criteria					
mLNR < 0.2	4	1.976 (1.425, 2.740)	0.056	2.878 (1.401, 5.912)	0.061
mLNR ≤ 0.2	7	2.131 (1.667, 2.723)	0.081	2.497 (1.706, 3.657)	0.015
mLNR ≥ 0.2	7	2.287 (1.905, 2.747)	0.324	2.322 (1.885, 2.861)	0.160
Colon	4	2.496 (1.917, 3.249)	0.495	2.496 (1.917, 3.249)	0.490
Rectum	2	2.158 (1.645, 2.831)	0.060	2.730 (1.226, 6.076)	NA

CI, confidence interval; mLNR, metastatic lymph node ratio; NA, not applicable. Colon cancer represent mLNR > 0.2, and rectal cancer present mLNR = 0.2, respectively.

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
