# Peer review of "Metastatic Lymph Node Ratio (mLNR) is a Useful Parameter in the Prognosis of Colorectal Cancer; A Meta-Analysis for the Prognostic Role of mLNR"

_medicina, 2019, doi:10.3390/medicina55100673_

Round 1
Reviewer 1 Report
Authors have replied all queries adequately.
Reviewer 2 Report
The authors have chosen to respond to the issues I raised in my first reviewing. They responded well to all criticisms and their corrections significantly improved the manuscript.
For reviewing round, the authors have added several elements such as Forest plots that significantly improve the presentation of the results.
I recommend to accept the manuscript in its present form.
Reviewer 3 Report
The authors have addressed the previous concerns raised by reviewers and also have presented the limitations of the study. I have no further comments.
This manuscript is a resubmission of an earlier submission. The following is a list of the peer review reports and author responses from that submission.
Round 1
Reviewer 1 Report
This study by Pyo et al presents an interesting manuscript studying Metastatic Lymph Node Ratio (mLNR) as a useful parameter in the Prognosis of Colorectal Cancer.
The study was well designed and the major strength of the study was large number of dataset.
The study included correlative data based on the tumor stage, location.
The study should also include values for age group, overall and disease-free follow-up times.
Moreover, the study could also be strengthened by correlating properties like vascular, perineural, lymphatic, and serosal invasions with the MLNR group.
Previous studies have indicated that mLNR ratio in various cancers including colorectal is a valuable prognostic factor, so this study although important only adds on to our existing knowledge.
Moreover, mLNR could be used as important prognostic factor independent of LN as 12 as few studies have indicated that and the authors rightfully state that as limitations of current study. Overall the study was well designed but could add valuable information as I mentioned above.
Minor comments: some of the sentences are grammatically incorrect which needs to be corrected
Example: The assessment for LN metastasis in CRC is performing via several affected regional LN in the 38 th AJCC cancer staging
Author Response
This study by Pyo et al presents an interesting manuscript studying Metastatic Lymph Node Ratio (mLNR) as a useful parameter in the Prognosis of Colorectal Cancer.
Point 1: The study was well designed and the major strength of the study was large number of dataset.
The study included correlative data based on the tumor stage, location.
The study should also include values for age group, overall and disease-free follow-up times.
Response 1:
We added the information for age in Table 1.
To avoid selection bias introduced by different follow-up periods, extraction after a 60-month follow-up period was conducted in the present study. Therefore, the information for the follow-up period could not be described.
Point 2: Moreover, the study could also be strengthened by correlating properties like vascular, perineural, lymphatic, and serosal invasions with the MLNR group.
Response 2:
The value of mLNR can be higher in cases with aggressive behaviors than in cases with indolent behaviors. However, in the present study, articles, which were studied the prognostic implication of mLNR, included. To assess the correlation between mLNR and aggressive behaviors, a new meta-analysis will be performed. We added the comment for this limitation in the revised manuscript as below:
Sixth, the value of mLNR can be higher in cases with aggressive behaviors than in cases with indolent behaviors. However, the detailed analysis could not be performed due to insufficient information on eligible studies.
Point 3: Previous studies have indicated that mLNR ratio in various cancers including colorectal is a valuable prognostic factor, so this study although important only adds on to our existing knowledge.
Moreover, mLNR could be used as important prognostic factor independent of LN as 12 as few studies have indicated that and the authors rightfully state that as limitations of current study. Overall the study was well designed but could add valuable information as I mentioned above.
Response 3:
Eligible studies had no information for the prognostic implication of mLNR according to the number of harvested LN. Thus, the impact of harvested LN on the prognostic implication could not be investigated in the present study.
Point 4: Minor comments: some of the sentences are grammatically incorrect which needs to be corrected
Example: The assessment for LN metastasis in CRC is performing via several affected regional LN in the 8th AJCC cancer staging
Response 4:
We corrected typo-errors and grammatical errors.
Reviewer 2 Report
First, I would like to thank you for giving me the opportunity to review this manuscript.
English is not my first language and I am not able to correct grammar mistakes (if present).
The aim of the study was to use a meta-analysis in order to determine the usefulness of mLNR for prediction of prognosis in CRC. The authors show that mLNR might be a useful prognostic factor.
I have some specific comments :
Introduction : the objectives of the study is well written.
Materials and methods
2.1. Searching studies and selection criteria -> the search terms should be more accurate if the authors precise the full research details instead of the terms used in the search box.
Table 1 : This table should be more detailled.
Results : the result section is well written.
Discussion : some of the limits of the study are not discussed, the authors should discuss other limitations of the study.
I have some general comments
One of the major issue of this study is to ahve included rectal cancer, because colon cancer and rectal cancer do not have the same treatment and prognosis.
An issue of this article, is that one the analyzed studies has much more patients than other studies (>200k patients), so the weigh of this study in the final analysiss in not discussed.
Another issue, is that little is known about the treatment of patient, or the microsatellite status. This is a major issue in an meta-analysis of the prognosis.
Author Response
First, I would like to thank you for giving me the opportunity to review this manuscript.
English is not my first language and I am not able to correct grammar mistakes (if present).
The aim of the study was to use a meta-analysis in order to determine the usefulness of mLNR for prediction of prognosis in CRC. The authors show that mLNR might be a useful prognostic factor.
I have some specific comments :
Introduction: the objectives of the study is well written.
Response:
Thank you for your careful review.
Materials and methods
2.1. Searching studies and selection criteria -> the search terms should be more accurate if the authors precise the full research details instead of the terms used in the search box.
Response:
We corrected the searching term in the method part as below:
“colon OR rectum OR colorectal” AND “lymph node ratio OR metastatic lymph node ratio”
Table 1: This table should be more detailed.
Response:
We added detailed information, including age, harvested number of LN, and the number of patients with high and low mLNR.
Results: the result section is well written.
Response:
Thank you for your careful review.
Discussion: some of the limits of the study are not discussed, the authors should discuss other limitations of the study.
Response:
We added the limitation as below:
Sixth, the value of mLNR can be higher in cases with aggressive behaviors than in cases with indolent behaviors. In addition, the correlation between mLNR and survival according to MSI status is unclear. However, the detailed analysis could not be performed due to insufficient information on eligible studies.
I have some general comments
One of the major issue of this study is to have included rectal cancer, because colon cancer and rectal cancer do not have the same treatment and prognosis.
Response:
We added the subgroup analysis by tumor location for cut-off of high mLNR and showed the results in Table 2 and 3.
An issue of this article, is that one the analyzed studies has much more patients than other studies (>200k patients), so the weigh of this study in the final analysis in not discussed.
Response:
The impact of Zhang’s report (2018) might be larger than other eligible studies. The impact of the individual study was evaluated by a sensitivity analysis. In omitting Zhang’s report, the estimated HR was 1.834 (95% CI 1.391-2.419). Compared to overall cases (HR 1.617, 95% CI 1.393-1.877), the tendency did not differ. We added this result in the revised manuscript.
Another issue, is that little is known about the treatment of patient, or the microsatellite status. This is a major issue in an meta-analysis of the prognosis.
Response:
Eligible studies had no information between MSI and prognosis. Thus, this point did not discuss in the current study. We added the limitation as below:
Sixth, the value of mLNR can be higher in cases with aggressive behaviors than in cases with indolent behaviors. In addition, the correlation between mLNR and survival according to MSI status is unclear. However, the detailed analysis could not be performed due to insufficient information on eligible studies.
Reviewer 3 Report
The method for analysis is conventional. The number of involved cases/reports is so huge that this article will contribute to the researchers and clinical doctors to some extent. It is agreeable that the authors assessed their result appropriately and that they suggested what they will need next.
Author Response
The method for analysis is conventional. The number of involved cases/reports is so huge that this article will contribute to the researchers and clinical doctors to some extent. It is agreeable that the authors assessed their result appropriately and that they suggested what they will need next.
Response:
Thank you for your careful review.
Reviewer 4 Report
Authors conducted a meta-analysis of the correlation between mLNR and survival rate from 14 eligible studies of 225,607 patients by searching the PubMed and MEDLINE databases. The results indicated that a high mLNR had significant correlation with worse overall survival and disease-free survival rates in CRC patients. Regarding tumor location, rectal cancer showed a worse survival rate when compared to colon cancer. In the analysis for overall survival, when mLNR was 0.2, HR was the highest across the different subgroups. However, in the analysis for disease-free survival, the subgroup with an mLNR < 0.2 had a higher HR than the other subgroups. Taken together, mLNR may be a useful prognostic factor for patients with CRC, regardless of the tumor stage or tumor location. The results seems informative and appealing; however, there are a lot of criticisms and have several issues that the authors need to address before the manuscript is suitable for publication.
Major Compulsory Revisions:
1. The major flow of the current study was that 14 eligible studies should be further elucidated as some studies were colon or colorectal or rectal cancer patients. The retrieval of lymph node would be largely dependent the tumor location. The different distribution of lymph node retrieval needs to be clarified in tumor location. Another major concern was that not all patients were stage III disease, 3 studies did not show the stage III patients and the actual patients’ number of stage III is suggested to be listed in Table 1.
2. The mLNR is based on the harvested number of lymph node and the issue should be considered in the initial study design.
3. Another major flaw was the definition of mLNR varied among 14 studies, the actual cut-off point needs to be clarified. Furthermore, some studies with very limited case numbers and most of analyzed patients (96.7%) were belong to one study from USA.
4. The results showed that HR of subgroup included all stage tumors was 2.420, and it raised the question that for all stage? Stage I, II and IV, the real role of mLNR is questionable.
5. For overall survival of CRCs, HR of rectal cancer was higher than that of colon cancer in the current study. The stratification of CRC patients into colon and rectal cancer subgroup is mandatory.
6. Table 2 and 3 showed the confounding factors of tumor stage and tumor location on mLNR actually existed. Therefore, the forest plots analysis should replace the tables 2 and 3 to assure the results.
7. In Discussion section, authors focused on stage III only, it would remind of us that mLNR was on the stage III not for all stage.
Minor Essential Revisions:
1. Please correct the typo and grammatical error.
2. A funnel plot graph to check for the existence of publication bias is suggested to be added.
Author Response
Authors conducted a meta-analysis of the correlation between mLNR and survival rate from 14 eligible studies of 225,607 patients by searching the PubMed and MEDLINE databases. The results indicated that a high mLNR had significant correlation with worse overall survival and disease-free survival rates in CRC patients. Regarding tumor location, rectal cancer showed a worse survival rate when compared to colon cancer. In the analysis for overall survival, when mLNR was 0.2, HR was the highest across the different subgroups. However, in the analysis for disease-free survival, the subgroup with an mLNR < 0.2 had a higher HR than the other subgroups. Taken together, mLNR may be a useful prognostic factor for patients with CRC, regardless of the tumor stage or tumor location. The results seems informative and appealing; however, there are a lot of criticisms and have several issues that the authors need to address before the manuscript is suitable for publication.
Major Compulsory Revisions:
Point 1: The major flow of the current study was that 14 eligible studies should be further elucidated as some studies were colon or colorectal or rectal cancer patients. The retrieval of lymph node would be largely dependent the tumor location. The different distribution of lymph node retrieval needs to be clarified in tumor location. Another major concern was that not all patients were stage III disease, 3 studies did not show the stage III patients and the actual patients’ number of stage III is suggested to be listed in Table 1.
Response 1:
As recommended by a reviewer, the estimated mean number of harvested lymph nodes was investigated according to tumor location. The estimated mean number of harvested lymph nodes was 23.365 (95% CI 20.993-25.736) in overall cases. In subgroups by tumor location, the estimated mean number of harvested lymph nodes was 27.709 (95% CI 24.435-30.983) and 15.511 (95% CI 10.219-20.803) in colon and rectum, respectively. The harvested number of lymph nodes was significantly higher in the colon than in rectum (P = 0.001 in a meta-regression test). We added these results in the revised manuscript. Also, the method for a meta-regression test was described in the revised manuscript.
In addition, the detailed information for the patients’ number of stage III added in Table 1.
Point 2: The mLNR is based on the harvested number of lymph node and the issue should be considered in the initial study design.
Response 2:
This study aimed to elucidate the prognostic role of the metastatic lymph node ratio (mLNR) in patients with colorectal cancer (CRC), using a meta-analysis. As described in the manuscript, the number of metastatic lymph node can be associated with the harvested number of lymph nodes. To diminish the impact of harvested lymph nodes, mLNR has been studied as the complementary parameter. However, unlike the conventional retrospective study, the meta-analysis can’t be handled two independent variables (mLNR & harvested lymph node).
Point 3: Another major flaw was the definition of mLNR varied among 14 studies, the actual cut-off point needs to be clarified. Furthermore, some studies with very limited case numbers and most of analyzed patients (96.7%) were belong to one study from USA.
Response 3:
We described the actual cut-off point of eligible studies in Table 1.
In addition, the impact of Zhang’s report (2018) might be larger than other eligible studies. The impact of the individual study was evaluated by a sensitivity analysis. In omitting Zhang’s report, the estimated HR was 1.834 (95% CI 1.391-2.419). Compared to overall cases (HR 1.617, 95% CI 1.393-1.877), the tendency did not differ.
Point 4: The results showed that HR of subgroup included all stage tumors was 2.420, and it raised the question that for all stage? Stage I, II and IV, the real role of mLNR is questionable.
Response 4:
In the TNM staging system, patients with stage I or II had no lymph node metastasis. Because mLNR of these cases was 0, the impact of mLNR can’t be evaluated in patients with stage I or II. In the present study, these cases were excluded because of the comparison of cases with lymph node metastasis.
Stage IV subgroup has distant metastases through lymphovascular invasion, included cases with and without lymph node metastasis. The impact of mLNR in predicting the prognosis might be limited. However, among eligible studies, there was no information for the comparison between high and low mLNR in stage IV.
Point 5: For overall survival of CRCs, HR of rectal cancer was higher than that of colon cancer in the current study. The stratification of CRC patients into colon and rectal cancer subgroup is mandatory.
Response 5:
In rectum subgroup, cut-offs for high mLNR were 0.2 and 0.25 (Leonard 2016 and Park 2015, respectively). Among the two studies, only one report showed the information for the overall survival (Leonard 2016). In mLNR ≥ 0.2 subgroup, HRs of the colon and rectal cancers were 2.412 (95% CI 1.463-3.977) and 5.040 (95% CI 1.780-14.270), respectively. This result was same as mLNR = 0.2 vs. mLNR > 0.2. As recommended by a reviewer, the title was replaced to colon and rectum in Table 2.
In addition, data for disease-free survival was corrected by tumor location in Table 3.
Point 6: Table 2 and 3 showed the confounding factors of tumor stage and tumor location on mLNR actually existed. Therefore, the forest plots analysis should replace the tables 2 and 3 to assure the results.
Response 6:
Forest plots (Figure 2) were deleted in the revised version. The result of Fig. 2 included in Table 2 and 3. In addition, less critical data was deleted in Table 2 and 3.
Point 7: In Discussion section, authors focused on stage III only, it would remind of us that mLNR was on the stage III not for all stage.
Response 7:
In the TNM staging system, patients with stage I or II had no lymph node metastasis. Because mLNR of these cases was 0, the impact of mLNR can’t be evaluated in patients with stage I or II. In the present study, these cases were excluded because of the comparison of cases with lymph node metastasis. Thus, stage III rather than other stages was focusing in the present study.
Minor Essential Revisions:
1. Please correct the typo and grammatical error.
Response:
We corrected typo-errors and grammatical errors.
2. A funnel plot graph to check for the existence of publication bias is suggested to be added.
Response:
Publication bias is assessed through a funnel plot, Egger’s test, fail-safe N test, and trim-fill test. Statistical analysis for a funnel plot is Egger’s test and shows using P-value. In the present study, because too many funnel plots are needed for various subgroup analysis, P-value from Egger’s test used to assess the publication bias.
Round 2
Reviewer 2 Report
The authors have chosen to respond to the issues I raised in my first reviewing. They responded well to all criticisms and their corrections significantly improved the manuscript.
I recommend to accept the manuscript in its present form.
Author Response
The authors have chosen to respond to the issues I raised in my first reviewing. They responded well to all criticisms and their corrections significantly improved the manuscript.
I recommend to accept the manuscript in its present form.
Response:
Thank you for your careful review.
Reviewer 4 Report
Authors replied all the raised questions except for the following 2 queries:
Point 5: For overall survival of CRCs, HR of rectal cancer was higher than that of colon cancer in the current study. The stratification of CRC patients into colon and rectal cancer subgroup is mandatory.
Response 5:
In rectum subgroup, cut-offs for high mLNR were 0.2 and 0.25 (Leonard 2016 and Park 2015, respectively). Among the two studies, only one report showed the information for the overall survival (Leonard 2016). In mLNR ≥ 0.2 subgroup, HRs of the colon and rectal cancers were 2.412 (95% CI 1.463-3.977) and 5.040 (95% CI 1.780-14.270), respectively. This result was same as mLNR = 0.2 vs. mLNR > 0.2. As recommended by a reviewer, the title was replaced to colon and rectum in Table 2.
In addition, data for disease-free survival was corrected by tumor location in Table 3.
Query: Table 3 was confusing. Colon and rectum was categorized into High mLNR criteria?
Point 6: Table 2 and 3 showed the confounding factors of tumor stage and tumor location on mLNR actually existed. Therefore, the forest plots analysis should replace the tables 2 and 3 to assure the results.
Response 6:
Forest plots (Figure 2) were deleted in the revised version. The result of Fig. 2 included in Table 2 and 3. In addition, less critical data was deleted in Table 2 and 3.
Query: Authors misunderstand the original criticism. I recommended authors to present with the Forest plot to replace the Tables 2 and 3, of which Forest plot will be easier for readers. Please ask an expert good at statistical analysis to perform it.
Author Response
Authors replied all the raised questions except for the following 2 queries:
Point 5: For overall survival of CRCs, HR of rectal cancer was higher than that of colon cancer in the current study. The stratification of CRC patients into colon and rectal cancer subgroup is mandatory.
Response 5:
In rectum subgroup, cut-offs for high mLNR were 0.2 and 0.25 (Leonard 2016 and Park 2015, respectively). Among the two studies, only one report showed the information for the overall survival (Leonard 2016). In mLNR ≥ 0.2 subgroup, HRs of the colon and rectal cancers were 2.412 (95% CI 1.463-3.977) and 5.040 (95% CI 1.780-14.270), respectively. This result was same as mLNR = 0.2 vs. mLNR > 0.2. As recommended by a reviewer, the title was replaced to colon and rectum in Table 2.
In addition, data for disease-free survival was corrected by tumor location in Table 3.
Query: Table 3 was confusing. Colon and rectum was categorized into High mLNR criteria?
Response: As we described in response 5, HRs of the colon cancer and mLNR > 0.2 were same and HRs of rectal cancer and mLNR = 0.2 were also same.
To avoid the confusion, we added footnotes in Table 2 and Table 3.
Point 6: Table 2 and 3 showed the confounding factors of tumor stage and tumor location on mLNR actually existed. Therefore, the forest plots analysis should replace the tables 2 and 3 to assure the results.
Response 6:
Forest plots (Figure 2) were deleted in the revised version. The result of Fig. 2 included in Table 2 and 3. In addition, less critical data was deleted in Table 2 and 3.
Query: Authors misunderstand the original criticism. I recommended authors to present with the Forest plot to replace the Tables 2 and 3, of which Forest plot will be easier for readers. Please ask an expert good at statistical analysis to perform it.
Response: As the reviewer has mentioned, we added new Forest plots (Figure 3) for the subgroup analysis by the tumor location in overall survival and disease-free survivals, respectively.
We also removed duplicated data and modified Tables.
We thank the reviewers for the comments and appreciate the opportunity to improve the manuscript.